# Impact of Diabetes Mellitus on Outcomes after High-Risk Interventional Coronary Procedures

**DOI:** 10.3390/jcm9113414

**Published:** 2020-10-25

**Authors:** Laura Johannsen, Julian Soldat, Andrea Krueger, Amir A. Mahabadi, Iryna Dykun, Matthias Totzeck, Rolf Alexander Jánosi, Tienush Rassaf, Fadi Al-Rashid

**Affiliations:** Department of Cardiology and Vascular Medicine, West German Heart and Vascular Center Essen, University Hospital Essen, Medical Faculty, 45147 Essen, Germany; Laura.Johannsen@uk-essen.de (L.J.); juliansoldat@googlemail.com (J.S.); andrea.krueger.68@stud.uni-due.de (A.K.); Amir-Abbas.Mahabadi@uk-essen.de (A.A.M.); Iryna.Dykun@uk-essen.de (I.D.); Matthias.Totzeck@uk-essen.de (M.T.); Alexander.Janosi@uk-essen.de (R.A.J.); Tienush.Rassaf@uk-essen.de (T.R.)

**Keywords:** high-risk percutaneous coronary intervention, Impella, mechanical circulatory support, diabetes mellitus

## Abstract

An increasing number of patients with coronary artery disease are at high operative risk due to advanced age, severe comorbidities, complex coronary anatomy, and reduced ejection fraction. Consequently, these high-risk patients are often offered percutaneous coronary intervention (PCI) as an alternative to coronary artery bypass grafting (CABG). We aimed to investigate the outcome of patients with diabetes mellitus (DM) undergoing high-risk PCI. We analyzed consecutive patients undergoing high-risk PCI (period 01/2016–08/2018). In-hospital major adverse cardiac and cerebrovascular events (MACCEs), defined as in-hospital stroke, myocardial infarction and death, and the one-year incidence of death from any cause were assessed in patients with and without DM. There were 276 patients (age 70 years, 74% male) who underwent high-risk PCI. Eighty-six patients (31%) presented with DM (insulin-dependent DM: *n* = 24; non-insulin-dependent DM: *n* = 62). In-hospital MACCEs occurred in 9 patients (3%) with a non-significant higher rate in patients with DM (*n* = 5/86, 6% vs. *n* = 4/190 2%; *p* = 0.24). In patients without DM, the survival rate was insignificantly higher than in patients with DM (93.6% vs. 87.1%; *p* = 0.07). One-year survival was not significantly different in DM patients with more complex coronary artery disease (SYNTAX I-score ≤ 22: 89.3% vs. > 22: 84.5%; *p* = 0.51). In selected high-risk patients undergoing high-risk PCI, DM was not associated with an increased incidence of in-hospital MACCEs or a decreased one-year survival rate.

## 1. Introduction

Diabetes mellitus (DM) is associated with a wide range of complications and affects the morbidity and mortality associated with cardiovascular diseases and procedures [1]. The pattern of coronary artery disease (CAD) in diabetic patients is often complex, with multiple lesions and widespread disease [2], making it difficult to achieve complete revascularization and adversely affecting the long-term prognosis [3,4,5]. According to the current European Society of Cardiology (ESC) guidelines on myocardial revascularization, coronary artery bypass grafting (CABG) should be favored over percutaneous coronary intervention (PCI) in patients with DM without contraindications [6]. The current guidelines recommend that decisions about PCI or CABG should be guided by the anatomical synergy between PCI with taxus and cardiac surgery (SYNTAX) score [6]. Risk stratification by SYNTAX score in patients with multivessel disease or left main coronary artery disease has been reported [7,8,9]. However, an increasing number of patients are at high operative risk due to advanced age, severe comorbidities, and reduced left ventricular ejection fraction [10,11]. In these patients, cardiac surgery is often refused. Consequently, these high-risk patients are currently often offered PCI as an alternative to CABG [10,12]. We recently defined a setting for the performance of safe and efficient high-risk interventional (HRI) procedures using an algorithm that incorporates anatomical complexity, comorbidity, and clinical presentation [11]. The aim of this study was to investigate and compare the outcomes of patients with and without DM undergoing high-risk PCI in this validated setting.

## 2. Material and Methods

We included consecutive patients who underwent high-risk PCI from January 2016 to August 2018 at our tertiary care center. The decision on high-risk PCI was based on a “Heart Team-based” NOVA-HRI algorithm [11]. Based on this algorithm, patients were categorized into 3 risk groups (I–III) with a stepwise increase in periprocedural management. The risk classification in this algorithm based on anatomical complexity (as defined by the SYNTAX I score), severe comorbidities (chronic obstructive pulmonary disease, severe aortic valve stenosis III°, carotid artery disease, chronic kidney disease stage ≥ 4, severe pulmonary hypertension, peripheral artery disease stage 4, stroke within 30 days prior to PCI, active infection/sepsis, and cancer with concurrent cancer therapy), and the current clinical presentation (left-ventricular ejection fraction) [11]. Patients with cardiogenic shock and those presenting with ongoing cardiopulmonary resuscitation (prior to coronary angiography) were excluded from the study.

The study was approved by the institutional ethics committee of the University of Duisburg-Essen (Essen, Germany—19-8562-BO). All procedures were performed in accordance with relevant guidelines and regulations [6,13,14].

Data on the laboratory values, risk factors and clinical diagnoses of patients were obtained from all available hospital records. In elective cases, patients underwent transthoracic echocardiography (TTE). In ACS patients, TTE was performed in advance of PCI. All further examinations were performed by clinical indication based on symptomatic or physical examination. In cases of stable coronary artery disease, complete revascularization, defined as a residual SYNTAX I score ≤ 8 [15], was the goal. Patients presenting with ST-segment elevation myocardial infarction underwent PCI for the culprit lesion only during the index procedure, followed by complete coronary revascularization of nonculprit lesions if required. Patients with unstable angina and non-ST-segment elevation myocardial infarction were completely revascularized during the HRI procedure. Only second-generation drug-eluting stents were implanted.

We stratified each cohort by diabetes status: DM and no DM. Patients were defined as having DM if they had previously been hospitalized for a diabetes diagnosis or were receiving treatment with diet, antidiabetic oral drugs, or insulin. For further analysis, patients with DM were divided into two groups: insulin-dependent DM (IDDM) and non-insulin-dependent DM (NIDDM).

The primary outcome was one-year survival of patients undergoing high-risk interventions. The secondary outcomes were (i) in-hospital major adverse cardiac and cerebrovascular events (MACCEs), defined as in-hospital stroke, myocardial infarction [16], and death (until 30 days post PCI); and (ii) MACCEs and one-year survival in DM patients with respect to angiographic disease complexity, defined by the SYNTAX I score [17].

Acute kidney injury was defined as stage ≥ 1 according to the KDGIO definition [18]. Vascular complications were defined as access site- or access-related vascular injury, as defined by the Valve Academic Research Consortium-2 [19]. Coronary complications were defined as acute stent thrombosis, dissection, no reflow, or coronary perforation [20].

Data are presented as means ± standard deviations if normally distributed or as the medians and interquartile ranges otherwise. Categorical variables are presented as the frequencies and percentages. Categorical data were compared between groups using the χ^2^ or Fisher’s exact tests. Continuous variables were compared using Student’s t-test if normally distributed or the Mann–Whitney U test if not. Survival curves were constructed using Kaplan–Meier estimates based on all available data and were compared with a log-rank test. A *p*-value < 0.05 indicated statistical significance. All analyses were performed using SPSS software (version 25, SPSSS, Chicago, IL, USA).

## 3. Results

### 3.1. Baseline and Procedural Characteristics

A total of 276 patients (mean age 70 years, 74% male) underwent high-risk PCI (Table 1). Eighty-six patients (31%) presented with DM (IDDM: *n* = 24; NIDDM: *n* = 62). Patients with DM were more obese (BMI 29 ± 5 vs. 27 ± 5 kg/m^2^; *p* = 0.001), and the rates of atrial fibrillation (27% vs. 16%; *p* = 0.05) and arterial hypertension (94% vs. 81%; *p* = 0.03) was significantly higher in DM patients than in non-DM patients. The baseline logistic EuroSCORE and SYNTAX I scores were not significantly different. PCI procedures with mechanical circulatory support were performed with no significantly difference in both groups (DM: *n* = 22 (26%); no DM: *n* = 39 (21%); *p* = 0.35). In most cases, (90%), a complete revascularization could be achieved. There were no significant differences regarding the complexity of PCI, the amount of contrast agent and the usage of mechanical circulatory support systems between these groups (Table 2).

### 3.2. No Significant Difference between Patients with and without Diabetes Mellitus

In-hospital MACCEs occurred in 3% of all patients (*n* = 9). There were no significant differences (*p* = 0.24) between patients with and without DM (Table 3). Postprocedural acute kidney injury was more frequent in patients with DM than in those without DM (16% vs. 6%; *p* = 0.02). There was also no significant difference in the occurrence of vascular and coronary complications (Table 3). Kaplan–Meier survival curves (Figure 1a) revealed a higher survival rate in patients without DM than in those with DM (93.6% vs. 87.1%). However, this difference was not significant at one year (log-rank *p* = 0.07). One-year survival of patients with NDDM compared with patients with IDDM did not significantly differ (90.2 vs. 78.9; log-rank *p* = 0.17; Figure 1b).

### 3.3. Complex CAD Had No Influence on In-Hospital MACCE

For further analysis of DM patients, we focused on the outcome based on angiographic disease complexity, as defined by the SYNTAX I score. Patients with more complex CAD, defined by a SYNTAX I score > 22, did not have a higher rate of in-hospital MACCEs than those with less complex CAD (Figure 2a). One-year survival was also not different in DM patients with more and less complex CAD (89.3 vs. 84.5%; log-rank *p* = 0.51; Figure 2b).

## 4. Discussion

The findings of our study indicate that performing high-risk PCI in patients with DM was not associated with increased in-hospital MACCEs, even in patients with high baseline SYNTAX score. We also found that DM was associated with a nonsignificant decrease in one-year survival, which was mainly driven by an increased overall mortality in the IDDM subgroup.

The latest revascularization guidelines recommend CABG for subgroups such as diabetic patients with multivessel disease (MVD) and high SYNTAX scores [6,21]. This is due to the much more aggressive disease course in the smaller coronary vessels with a more diffuse distribution, making CABG the preferred treatment option, especially in patients with DM [22]. However, an increasing number of patients are at high operative risk due to advanced age, severe comorbidities, and reduced left ventricular ejection fraction [11]. Given to the rising of high-risk diabetic patients, we are witnessing an increase of the high-risk percutaneous coronary interventions (PCI) in this population [10].

The baseline characteristics of our study patients indicate a high-risk cohort with high anatomical complexity with high baseline SYNTAX I scores [23,24]. Compared to the SYNTAX II study our cohort had a similar severity of CAD but a lower left ventricular ejection fraction and a higher rate of comorbidities [25]. In a subgroup analysis of the SYNTAX study in 452 patients with DM and 3-vessel CAD or baseline stenosis both at 1 year (CABG: 14.2% vs. PCI: 26%, *p* = 0.003), a significantly lower MACCE rate was found in the CABG group compared to the PCI group [26]. However, taking into account the SYNTAX score, only those diabetic patients with complex disease (SYNTAX score > 33) benefited from CABG treatment, whereas in patients with DM and mild or intermediate complexity CHD (SYNTAX score < 22 or 22–32), this advantage of CABG over PCI was no longer detectable [17]. Significant limitations of the SYNTAX trial, in contrast to the current clinical standard of care, were the exclusive use of bare-metal stents or first-generation drug-coated stents and, for diabetic patients, the small study population [27].

Based on this, the FREEDOM (Future Revascularization Evaluation in Patient with Diabetes Mellitus: Optimal Management of Multivessel Disease) trial included 1900 patients with MVD and DM, which were randomized to either PCI with mostly first-generation drug-eluting stents or CABG [28]. Approximately one-third of the patients had recent acute coronary syndrome, and nearly two-thirds had high baseline SYNTAX I scores (26.2 ± 8.4% for PCI, 26.1 ± 8.8% for CABG). At two years, the primary composite outcome of all-cause mortality, nonfatal myocardial infarction, and nonfatal stroke occurred in 13% of PCI patients and in 11.9% of CABG patients (*p* = 0.005). Both all-cause mortality (6.7 vs. 6.3%; *p* = 0.05) and nonfatal MI were reduced in the CABG group, although stroke was increased in the CABG group. It should be noted that with respect to the primary outcome, there was an early hazard of CABG, and survival curves separated in favor of CABG at approximately two years [28]. Our data are comparable, although our cohort is a higher risk cohort compared to that in the FREEDOM trial, with older patients (70 vs. 63 years), a higher baseline EuroSCORE (11% vs. 3%) and a lower left ventricular ejection fraction (46 vs. 66%). A significant limitation of the FREEDOM study was the use of first generation drug-eluting stents [29,30].

Interestingly the rate of patients with MACCE was low. In this number of patients, we did not detect a significant difference in in-hospital MACCEs or one-year survival between DM patients with low and intermediate/high levels of anatomical complexity (as defined by a SYNTAX score cut-off value of 22) who underwent PCI. Patients with IDDM had a decreased survival at one year (NIDDM 90.2 vs. IDDM 78.9; log-rank *p* = 0.17; Figure 1b). These results are in accordance with published data. Patients treated with insulin have worse outcomes compared to patients treated with oral hypoglycemics or diet in the setting of acute myocardial infarction [31], after PCI [32] and after CABG [33]. It is still debatable whether insulin directly affects the outcome or whether it is merely a marker of advanced diabetes. The present study has several limitations. The analysis is based on a single-center cohort of a limited number of patients with a small amount of MACCE. This is due to the fact that a very specific group of patients was examined in this study. We only investigated mortality up to one year and not the rate of repeat revascularization. Therefore, these data should be considered hypothesis-generating. The reproducibility of the SYNTAX score is a concern and can be profoundly influenced by the experience of scorers. Operator experience may be a potential confounder. Our observational data can only be used to report associations. Causal relations cannot be inferred due to the limited number of patients who meet the criteria for this study.

Finally, our high-risk cohort represents a growing population [34] with specific “tailored” indications for PCI, often due to a high operational risk during cardiac surgery. In patients with a balanced treatment option postulated in the Heart Team decision for both revascularization procedures, the presence of DM, the anatomical severity of CAD, and the presence of severe comorbidities should be among the key criteria used when selection optimal individual therapeutic procedures [10]. High-risk PCI was not associated with a significant increase in-hospital MACCE rate in high-risk patients with DM. There was also no significant difference in one-year survival, even in patients with high SYNTAX scores. Nonetheless, larger prospective randomized controlled multi-center trials with long-term follow-up results are mandatory to confirm and establish our findings. Until further trial data are available, high-risk PCI may offer an alternative treatment strategy for selected high-risk diabetic patients. In this predefined and validated setting PCI seems to be a safe treatment option for high-risk diabetic patients.

## Figures and Tables

**Figure 1 jcm-09-03414-f001:**
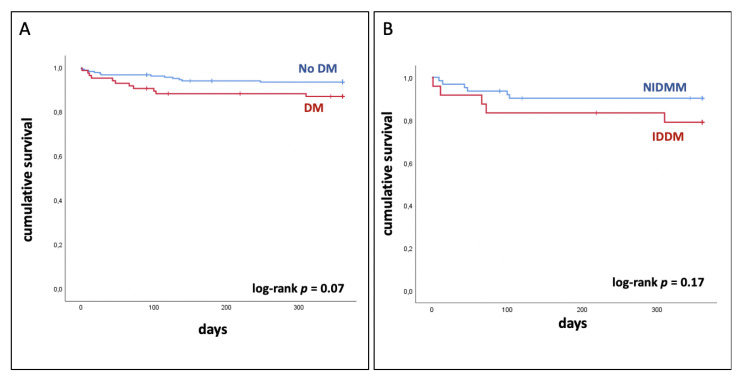
**Kaplan–Meier survival curves at one year** (**A**) Patients with and without diabetes mellitus (DM) who underwent high-risk percutaneous coronary intervention (PCI). (**B**) Survival curves only for DM patients; IDDM: insulin-dependent diabetes mellitus; NIDDM: non-insulin-dependent diabetes mellitus.

**Figure 2 jcm-09-03414-f002:**
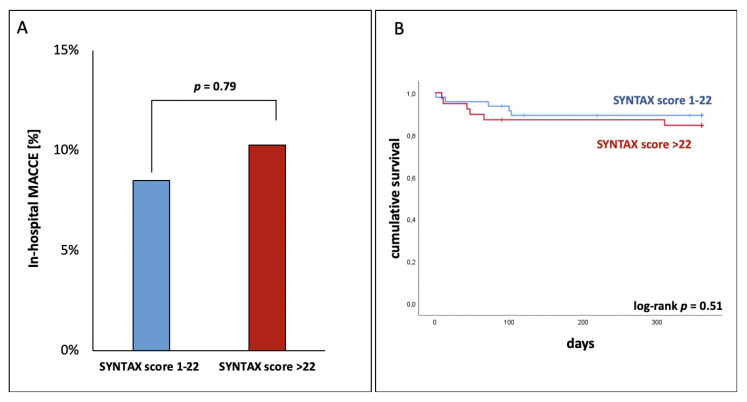
Association of coronary anatomical complexity with clinical outcomes in diabetic patients. (**A**) In-hospital MACCEs and (**B**) Kaplan–Meier estimates of one-year survival according to baseline SYNTAX I score; DM: diabetes mellitus; MACCE: major adverse cardiac and cerebrovascular events.

**Table 1 jcm-09-03414-t001:** Baseline Characteristics.

	All (*n* = 276)	No DM (*n* = 190)	DM (*n* = 86)	*p*-Value
age (yrs.), mean ± SD	70 ± 11	70 ± 11	70 ± 11	0.76
male sex, *n* (%)	203 (74)	138 (73)	65 (76)	0.66
body mass index (kg/m^2^), mean ± SD	27 ± 5	27 ± 5	29 ± 5	0.001
logistic EuroSCORE (%), mean ± SD	10 ± 12	9 ± 11	11 ±15	0.07
SYNTAX I score (%), mean ± SD	21 ± 10	21 ± 10	22 ± 11	0.31
LVEF (%), mean ± SD	47 ± 10	48 ± 10	46 ± 11	0.06
chronic obstructive pulmonary disease, *n* (%)	20 (7)	13 (7)	7 (8)	0.8
peripheral artery disease stage IV, *n* (%)	7 (3)	4 (2)	3 (4)	0.68
pulmonary hypertension, *n* (%)	36 (13)	24 (13)	12 (14)	0.85
CAD with prior PCI, *n* (%)	134 (49)	87 (46)	47 (55)	0.19
coronary artery bypass grafting, *n* (%)	31 (11)	22 (12)	9 (11)	0.84
prior cardiac surgery, *n* (%)	37 (13)	25 (13)	12 (14)	0.85
atrial fibrillation, *n* (%)	53 (19)	30 (16)	23 (27)	<0.05
hypertension, *n* (%)	234 (85)	153 (81)	81 (94)	0.03
baseline creatinine (mg/dL), mean ± SD	1.32 ± 0.84	1.27 ± 0.82	1.45 ± 0.86	0.09

CAD: coronary artery disease; DM: diabetes mellitus; LVEF: left-ventricular ejection fraction; PCI: percutaneous coronary intervention; SYNTAX: synergy between percutaneous coronary intervention with taxus and cardiac surgery.

**Table 2 jcm-09-03414-t002:** Procedural Data.

	All (*n* = 276)	No DM (*n* = 190)	DM (*n* = 86)	*p*-Value
acute coronary syndrome, *n* (%)	129 (46)	86 (45)	43 (50)	0.52
MCS, *n* (%)	61 (22)	38 (20)	23 (27)	0.25
multivessel PCI	265 (96)	181 (95)	84 (98)	0.51
residual SYNTAX I score < 8%, *n* (%)	247 (90)	170 (90)	77 (90)	0.98
PCI left main artery, *n* (%)	103 (37)	65 (34)	38 (44)	0.14
PCI left anterior descending coronary artery, *n* (%)	169 (61)	112 (59)	57 (66)	0.29
PCI left circumflex coronary artery, *n* (%)	127 (46)	86 (45)	41 (48)	0.8
PCI right coronary artery, *n* (%)	81 (29)	60 (32)	21 (24)	0.25
PCI bypass graft, *n* (%)	15 (5)	11 (6)	4 (5)	0.78
last remaining vessel, *n* (%)	3 (1)	2 (1)	1 (1)	0.94
contrast agent (mL), mean ± SD	247 ± 114	249 ± 121	241 ± 101	0.61
number of implanted stents per patient, median (IQR)	2 (1–7)	2 (1–7)	2 (1–7)	0.37
stent length per patient (mm), median (IQR)	42 (9–143)	42 (9–143)	44 (9–111)	0.99

DM: diabetes mellitus; MCS: mechanical circulatory support; PCI: percutaneous coronary intervention.

**Table 3 jcm-09-03414-t003:** In-hospital major adverse cardiac and cerebrovascular (MACCEs) and adverse events.

	All (*n* = 276)	No DM (*n* = 190)	DM (*n* = 86)	*p*-Value
MACCE, *n* (%)	9 (3)	4 (2)	5 (6)	0.24
stroke, *n* (%)	1 (1)	0	1 (1)	0.31
new myocardial infarction, *n* (%)	0	0	0	
death, *n* (%)	8 (3)	4 (2)	4 (5)	0.26
acute kidney injury, *n* (%)	24 (9)	11 (6)	13 (16)	0.02
vascular complications, *n* (%)	13 (5)	9 (5)	4 (5)	0.98
coronary complications, *n* (%)	9 (3)	5 (3)	4 (5)	0.38

DM: diabetes mellitus; MACCE: major adverse cardiac and cerebrovascular events.

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
