# Peer review of "Impact of Diabetes Mellitus on Outcomes after High-Risk Interventional Coronary Procedures"

_jcm, 2020, doi:10.3390/jcm9113414_

Round 1
Reviewer 1 Report
Johannsen et al. compared the rate of major adverse cardiac and cerebrovascular events and the one-year incidence of death from any cause in patients with and without Diabete Mellitus undergoing high-risk PCI. Despite the paper is clear and well-written there are several criticisms:
1- Considering the high prevalence of Diabete Mellitus, only 86 patients were enrolled in Diabete Mellitus group. The sample size is not quite numerous to evaluate differences between groups.
2- Considering the ESC Guidelines, a comparison between PCI and CABG in these "High Risk" patients could be more advisable and PCI may offer an alternative treatment strategy for selected high-risk diabetic patients.
Author Response
Reviewer #1
Johannsen et al. compared the rate of major adverse cardiac and cerebrovascular events and the one-year incidence of death from any cause in patients with and without Diabetes Mellitus undergoing high-risk PCI. Despite the paper is clear and well-written there are several criticisms:
- Considering the high prevalence of Diabetes Mellitus, only 86 patients were enrolled in Diabetes Mellitus group. The sample size is not quite numerous to evaluate differences between groups.
Response 1: We thank the reviewer for this comment. This analysis was part of a prospective study investigating safety and efficacy of a novel risk-stratification algorithm for high-risk coronary procedures1. In this study (including patients from 07/2017–06/2018) 1189 patients were screened and only 150 patients met the criteria for high-risk intervention. Other 55 patients underwent coronary artery bypass surgery. It should therefore be taken into account that we are examining a very specific patient collective in this study. Criteria for high-risk intervention incorporated the anatomical lesion complexity (defined by the SYNTAX I score), comorbidities (oxygen-dependent chronic obstructive pulmonary disease, severe aortic valve stenosis III°, carotid artery disease, chronic kidney disease stage ≥4, severe pulmonary hypertension, peripheral artery disease stage 4, stroke within 30 days prior to PCI, active infection/sepsis and cancer with concurrent cancer therapy), and clinical presentation, including hemodynamic status, to identify patients at increased risk for coronary interventions.
In an aging society, it can be assumed that this patient collective will most likely grow in future. It should certainly be made clear that with this small number of patients it is primarily tendencies that can be worked out. However, this study can provide the basis for following studies in future. We have added this important limitation to our revised manuscript (page 6, lines 187-188 and page 7, lines 192-193 of the revised version of the manuscript and below):
Discussion
The present study has several limitations. The analysis is based on a single-center cohort of a limited number of patients with a small amount of MACCE. This is due to the fact that a very specific group of patients was examined in this study. We only investigated mortality up to one year and not the rate of repeat revascularization. Therefore, these data should be considered hypothesis-generating. The reproducibility of the SYNTAX score is a concern and can be profoundly influenced by the experience of scorers. Operator experience may be a potential confounder. Our observational data can only be used to report associations. Causal relations cannot be inferred due to the limited number of patients who meet the criteria for this study.
- Considering the ESC Guidelines, a comparison between PCI and CABG in these "High Risk" patients could be more advisable and PCI may offer an alternative treatment strategy for selected high-risk diabetic patients.
Response 2: We thank the reviewer for this remark. We fully agree with the reviewer and believe in the potential for future studies. With an average age of 70 years, a reduced left ventricular ejection fraction and high comorbidity rate of over 42% most of the included patients were too sick to be included in randomized studies. Accordingly, a randomized controlled study is elementary to be able to answer this question. However, these patient show up in our daily practice and are often at too high a risk for surgical therapy. Therefore, PCI may offer an alternative treatment strategy for selected high-risk diabetic patients. We revised the manuscript and have toned down the statement (page 7, lines 199-202 of the revised version of the manuscript and below):
Discussion
High-risk PCI was not associated with a significant increase in-hospital MACCE rate in high-risk patients with DM. There was also no significant difference in one-year survival, even in patients with high SYNTAX scores. Nonetheless, larger prospective randomized controlled multi-center trials with long-term follow-up results are mandatory to confirm and establish our findings. Until further trial data are available, high-risk PCI may offer an alternative treatment strategy for selected high-risk diabetic patients. In this predefined and validated setting PCI seems to be a safe treatment option for high-risk diabetic patients.
References
- Al-Rashid F, Totzeck M, Mahabadi AA, Johannsen L, Luedike P, Lind A, Krueger A, Kamler M, Kahlert P, Janosi RA, Heusch G and Rassaf T. Safety and efficacy of a novel algorithm to guide decision-making in high-risk interventional coronary procedures. Int J Cardiol. 2020;299:87-92.
Reviewer 2 Report
Thank you for the possibility for review the paper of German authors. The paper titled Impact of Diabetes Mellitus on Outcomes after High-2 Risk Interventional Coronary Procedures I read with great interest. Subject of the paper is important because DM is one of the nightmares of modern medicine and often occurred in patients treated by coronary invasive procedures. Text of manuscript is interested and has been written by good English. Figures are good quality, with clinical relevance. Conclusions are good represented in results section. Conclusion that in selected high-risk patients undergoing high-risk PCI, DM was not associated with an increased incidence of in-hospital MACCEs or a decreased one-year survival rate was not common in literature and have good clinical value. I don’t have any criticism to the paper at all.
Main aim of the authors was to investigate the outcome of patients with diabetes mellitus (DM) undergoing high-risk PCI. The subject itself is not especially new, however paper is complex with good follow up and in my opinion shoud be interested for the readers.
the paper has been well written and text is easy to understand.
Conclusions of the papaer are in connection to the results presented in the paper.
Author Response
Reviewer #2
Thank you for the possibility for review the paper of German authors. The paper titled Impact of Diabetes Mellitus on Outcomes after High-2 Risk Interventional Coronary Procedures I read with great interest. Subject of the paper is important because DM is one of the nightmares of modern medicine and often occurred in patients treated by coronary invasive procedures. Text of manuscript is interested and has been written by good English. Figures are good quality, with clinical relevance. Conclusions are good represented in results section. Conclusion that in selected high-risk patients undergoing high-risk PCI, DM was not associated with an increased incidence of in-hospital MACCEs or a decreased one-year survival rate was not common in literature and have good clinical value. I don’t have any criticism to the paper at all.
Main aim of the authors was to investigate the outcome of patients with diabetes mellitus (DM) undergoing high-risk PCI. The subject itself is not especially new, however paper is complex with good follow up and in my opinion should be interested for the readers. the paper has been well written and text is easy to understand. Conclusions of the paper are in connection to the results presented in the paper.
Response 1: We thank the reviewer for his/her positive comment.
Reviewer 3 Report
Even with the recommendations by guidelines, an increasing number of patients with diabetes mellitus (DM) are at high operative risk due to advanced age, severe comorbidities and reduced left ventricular ejection fraction, and these high-risk patients are currently often offered percutaneous coronary intervention (PCI) as an alternative to coronary artery bypass grafting. They analyzed 276 consecutive patients undergoing high-risk PCI. In-hospital major adverse cardiac and cerebrovascular events (MACCEs) occurred in 9 patients (3%) without differences between patients with and without DM. In patients without DM, the survival rate was insignificantly higher than in patients with DM (93.6% vs. 87.1%; p=0.07). They concluded that in selected high-risk patients undergoing high-risk PCI, DM was not associated with an increased incidence of in-hospital MACCEs or a decreased one-year survival rate.
Comment to the Author
Although this article is interesting, there are some concerns.
- Even with the retrospective and observational study, it would be inappropriate to conclude that DM was not associated with an increased incidence of in-hospital MACCEs or a decreased one-year survival rate, because the incidences were apparently numerically higher in DM patients. The sample size was too small to reach the conclusion. This is a critical limitation of the current study.
- In Abstract section, the authors mentioned that “In-hospital MACCEs occurred in 9 patients (3%) without differences between patients with (n=4/190 2%) and without DM (n=5/86, 6%, p=0.24). Judging from the sentence, the incidence was higher in patients without DM, and the result was different from the data in Table 3.
- Which kind of stents were used in the current study? The authors must mention it.
Author Response
Reviewer #3
Even with the recommendations by guidelines, an increasing number of patients with diabetes mellitus (DM) are at high operative risk due to advanced age, severe comorbidities and reduced left ventricular ejection fraction, and these high-risk patients are currently often offered percutaneous coronary intervention (PCI) as an alternative to coronary artery bypass grafting. They analyzed 276 consecutive patients undergoing high-risk PCI. In-hospital major adverse cardiac and cerebrovascular events (MACCEs) occurred in 9 patients (3%) without differences between patients with and without DM. In patients without DM, the survival rate was insignificantly higher than in patients with DM (93.6% vs. 87.1%; p=0.07). They concluded that in selected high-risk patients undergoing high-risk PCI, DM was not associated with an increased incidence of in-hospital MACCEs or a decreased one-year survival rate.
Comment to the Author
- Although this article is interesting, there are some concerns. Even with the retrospective and observational study, it would be inappropriate to conclude that DM was not associated with an increased incidence of in-hospital MACCEs or a decreased one-year survival rate, because the incidences were apparently numerically higher in DM patients. The sample size was too small to reach the conclusion. This is a critical limitation of the current study.
Response 1: We thank the reviewer for raising this point. In this study we examined a very specific patient collective. With an average age of 70 years, a reduced left ventricular ejection fraction and high comorbidity rate of over 42% most of the included patients were too sick to be included in randomized studies. In an aging society, it can be assumed that this patient collective will most likely grow in future. We agree with the reviewer that with this small number of patients it is primarily tendencies that can be worked out. We have adjusted this in the revised manuscript (page 6, lines 187-188 and page 7, lines 192-193 of the revised version of the manuscript and below):
Discussion
The present study has several limitations. The analysis is based on a single-center cohort of a limited number of patients with a small amount of MACCE. This is due to the fact that a very specific group of patients was examined in this study. We only investigated mortality up to one year and not the rate of repeat revascularization. Therefore, these data should be considered hypothesis-generating. The reproducibility of the SYNTAX score is a concern and can be profoundly influenced by the experience of scorers. Operator experience may be a potential confounder. Our observational data can only be used to report associations. Causal relations cannot be inferred due to the limited number of patients who meet the criteria for this study
- In Abstract section, the authors mentioned that “In-hospital MACCEs occurred in 9 patients (3%) without differences between patients with (n=4/190 2%) and without DM (n=5/86, 6%, p=0.24). Judging from the sentence, the incidence was higher in patients without DM, and the result was different from the data in Table 3.
Response 2: We thank the reviewer for this remark. We revised the manuscript according to the Reviewer`s suggestion (page 1, lines 22-25 of the revised version of the manuscript and below):
Abstract:
Eighty-six patients (31%) presented with DM (insulin-dependent DM: n=24; non-insulin-dependent DM: n=62). In-hospital MACCEs occurred in 9 patients (3%) with a non-significant higher rate in patients with DM (n=5/86, 6% vs n=4/190 2%; p=0.24). In patients without DM, the survival rate was insignificantly higher than in patients with DM (93.6% vs. 87.1%; p=0.07). One-year survival was not significantly different in DM patients with more complex coronary artery disease (SYNTAX-score≤22: 89.3% vs. >22: 84.5%; p=0.51).
- Which kind of stents were used in the current study? The authors must mention it
Response 2: This point is well taken. In our study only second-generation drug-eluting stents were implanted. We added this important information to our manuscript (page 2, line 74 of the revised version of the manuscript and below):
Methods:
Data on the laboratory values, risk factors and clinical diagnoses of patients were obtained from all available hospital records. In elective cases, patients underwent transthoracic echocardiography (TTE). In ACS patients, TTE was performed in advance of PCI. All further examinations were performed by clinical indication based on symptomatic or physical examination. In cases of stable coronary artery disease, complete revascularization, defined as a residual SYNTAX I score ≤ 8 [15], was the goal. Patients presenting with ST-segment elevation myocardial infarction underwent PCI for the culprit lesion only during the index procedure, followed by the deferred complete coronary revascularization of nonculprit lesions if required. Patients with unstable angina and non–ST-segment elevation myocardial infarction were completely revascularized during the HRI procedure. Only second-generation drug-eluting stents were implanted.
Round 2
Reviewer 3 Report
The authors adequately revised the manuscript. I have no additional comment.